# Polyploid Induction and Karyotype Analysis of *Dendrobium officinale*

**Yang Liu [1], Shan-De Duan [1], Yin Jia [1], Li-Hong Hao [1], Di-Ying Xiang [1], Duan-Fen Chen [1,\*] and Shan-Ce Niu [1,2,\*]**

1   College of Horticulture, Hebei Agricultural University, Baoding 071001, China
2   State Key Laboratory of North China Crop Improvement and Regulation, Hebei Agricultural University, Baoding 071001, China
\*   Correspondence: chenduanfen@163.com (D.-F.C.); yynsc@hebau.edu.cn (S.-C.N.)

**Abstract:** *Dendrobium officinale* Kimura et Migo is an orchid with both medicinal and edible values and a high economic value. The wild resources of *D. officinale* are in an endangered state. Compared with the wild *D. officinale*, cultivated *D. officinale* exhibits inferior quality and a low content of medicinal components. Polyploid induction is a conventional breeding tool for genome doubling of species, which can effectively increase the total amount of plant components to improve the medicinal efficacy of *D. officinale*. In this study, *D. officinale* tetraploids were generated by treating the protocorms with colchicine. Morphological observations showed that tetraploids exhibited decreased plant size and leaf shape index and increased stem diameter. Cytological observations showed that the polyploid plants had larger stomata and a lower number of stomata per unit area compared with normal plants. The highest stomata variation of 30.00% was observed when the plant was treated with 0.3% colchicine for 24 h. Chromosomal observations showed that treatment of plants with 0.2% colchicine for 48 h resulted in the highest tetraploid induction rate of 10.00%. A total of 10 tetraploids were successfully obtained by inducing plant protoplasts with colchicine. The number of diploid *D. officinale* chromosomes was 38 with a base number of 19, and the karyotype formula was $2n = 2x = 38 = 24m + 14sm$ with a karyotype asymmetry coefficient of 60.59%, belonging to type 2B. The number of tetraploid *D. officinale* chromosomes was 76 with a base number of 19, and the karyotype formula was $2n = 4x = 76 = 58m + 18sm$ with a karyotype asymmetry coefficient of 60.04%, belonging to type 2B. This study determined the optimal mutagenesis treatment based on the chromosome observation results, investigated the relationship between the phenotype and ploidy level, and generated the polyploid germplasm of *D. officinale*, thereby laying the foundation for the breeding of new *D. officinale* cultivars enriched with compounds of medicinal value.

**Keywords:** chromosome; colchicine; polyploid breeding

## 1. Introduction

*Dendrobium officinale*, a perennial herb of the genus *Dendrobium* belonging to the family Orchidaceae, possesses abundant polysaccharides, alkaloids, phenols, and amino acids, which confer it with a high medicinal value [1]. With improved living standards and healthcare awareness of people, the medicinal value of *D. officinale* is increasingly being recognized [2]. However, with the market supply exceeding the demand, wild resources are gradually being depleted because of over-harvesting [3], which has led breeders to develop synthetic varieties. Artificially produced *D. officinale* is poor quality and contains lower amounts of medicinal compounds than wild *D. officinale*. Therefore, attempts should be made to improve the content of medicinal ingredients of artificially produced *D. officinale* [4].

Polyploids are organisms with three or more complete sets of chromosomes in their somatic cells [5]. The increased number of chromosomes, additional genomic interactions, and genetic alterations affect epigenetic changes in various phenotypes and regulate gene expression [6], resulting in polyploid plants that differ from diploid plants in their traits [7].

Polyploid plants are typically characterized by enlarged organs [8], greater resistance to stress [9], increased secondary plant metabolites [10], and improved utilization of beneficial components [11]. Therefore, the yield of metabolites in plants can be increased by polyploidization and increasing the level of target compounds [12,13]. Various polyploidy induction pathways have been reported [14], and chemical methods are among the most widely used mutagenesis techniques [15]. Chemical mutagenesis is advantageous because it does not majorly change the plant genome, resulting in stable and heritable mutations. Colchicine and herbicides are the commonly used chemical mutagens [16]. The main mechanism of action of chemical mutagens involves the inhibition of spindle formation during cell division to prevent the replicated chromosomes from being pulled toward the poles and inhibit the division of cells into two daughter cells. This leads to the doubling of chromosomes to form polyploid cells, eventually resulting in polyploid plants. Thus, polyploidy can be achieved by interfering with meiosis to allow for doubling of chromosomes within the cells by using chemical mutagens such as colchicine.

Diverse induction materials are used in polyploid breeding, and exophytes are available from various sources. Plant parts, such as seeds, stem segments, and clumped shoots, can be used as induction materials for polyploidization. Through different mutagenesis methods, the mutagenization rate can be increased by using different combinations of the mutagen concentration and induction time. While studying polyploidization induction of *D. officinale*, Zhang [17] performed the induction of three types of explants, namely seed embryos, protocorms, and stem segments, by using the immersion method and medium addition method, and found that the effects of different induction methods using different explant materials varied greatly. Zhang et al. [18] used different concentrations of colchicine and different treatment times to soak *D. officinale* protocorms and cluster buds, and reported that soaking the explants in 0.09% colchicine for 24 h was the most optimal condition, resulting in an induction rate of 48%. Wu [19] used the medium addition method for the induction of protocorms and clumped shoots and reported that protocorms were more sensitive to colchicine than clumped shoots. The artificial induction of polyploids in different orchids is also particularly important [20]. Induced polyploids in the genus Cardoso et al. [21] can be used to obtain plants of compact size with increased floral longevity, higher number of flowers in bloom, and flowers with higher stiffness and more resistance to transport. Vilcherrez et al. [22] studied different types of *Cattleya hybrida* explants subjected to different exposure times and colchicine concentrations, and reported that the maximum number of polyploid plants were obtained from primary bulbs treated with 500–750 μM of colchicine for 18 h. In another study, most tetraploids were produced by treating the seeds of *Taraxacum* kok-saghyz [23] with 0.1% colchicine for 48 h after germination. The polyploid induction rate of the adventitious shoot segments of "Fengtou" ginger could reach 18% after treatment with colchicine at a concentration of 150 mg/L for 7 days [24]. The highest tetraploid induction rate of 11.53% was achieved when the germinating seeds of *Trachyspermum ammi* L. [25] were treated with 0.05% colchicine for 24 h. As per previous studies of polyploidization, the criteria for determining plant variation differ based on plant morphological differences, plant stomatal differences, and chromosomal numbers [17–19]. However, different determination criteria have yielded different rates of polyploidy, which necessitates extensive studies in the field. In this study, the variation rate and induction rate were determined on the basis of the stomatal differences of the treated plants and the chromosome doubling number, respectively.

In this study, different combinations of colchicine concentration and induction time were used to induce polyploidy in *D. officinale*, and materials from the proto-bulb period were used as explants. We determined the optimum treatment combinations for obtaining the highest mutation rate and the highest polyploid induction rate of *D. officinale*, and obtained the germplasm of tetraploid *D. officinale* plants. Furthermore, we obtained the karyotype data of diploid and tetraploid *D. officinale* plants by using the conventional squash karyotype analysis to verify their doubling characteristics. This study provides

technical support and germplasm resources for creating polyploid germplasm of *D. officinale* and further plant breeding research.

## 2. Materials and Methods

### 2.1. Plant Material

The materials in this study were obtained from the capsules of artificially pollinated *D. officinale*, which were planted at the College of Horticulture, Hebei Agricultural University.

### 2.2. Preparation of D. officinale Protocorms

The capsules were rinsed under running tap water for 30 min and then disinfected in 0.1% $HgCl_2$ for 15 min in an ultra-clean bench; they were rinsed again in sterile water and kept sterile. In the ultra-clean bench, the pods were cut open, and the seeds were sown into $1/2$ MS + 0.1 mg·L$^{-1}$ NAA + 30 g·L$^{-1}$ sucrose + 7 g·L$^{-1}$ agar (PH 5.8) medium. The seeds were spread evenly on the surface of the medium with an applicator stick and were allowed to germinate and expand in orchid culture bottles for almost 40 days to obtain *D. officinale* protocorms. The environmental conditions for tissue culture were: temperature $(25 \pm 2)$ °C, light intensity at 46 μmol·s$^{-1}$·m$^{-2}$, a photoperiod of 12 h.

### 2.3. Colchicine Configuration

Of note, 200 mg of colchicine (XiaSi Corporation, Beijing, China) was placed in a beaker and dissolved by adding a few drops of ethanol. After complete dissolution, a mother liquor was prepared at a concentration of 10%; it was filtered and sterilized using a disposable syringe and a sterile filter membrane with a pore size of 0.22 μm in an ultra-clean bench. Then, the mother liquor was aspirated into a sterile bottle and diluted with sterile water to prepare 0.1%, 0.2%, 0.3%, and 0.4% colchicine solutions (this concentration was determined by referring to the treatment concentration of previous studies [14–16]). The whole process was performed under aseptic conditions. The prepared solutions were stored in the dark.

### 2.4. Induction of Protocorms of D. officinale

The protocorms in good growth condition, uniform size, tender green color, and without differentiation were soaked in colchicine at the concentrations of 0.1%, 0.2%, 0.3%, and 0.4% for 24 h, 36 h, and 48 h, respectively. The soaked protocorms were protected from light and placed on a shaker to ensure that the material was in full contact with the colchicine solution. The plants were then repeatedly rinsed three to four times with sterile water on an ultra-clean bench, and the water was blotted out with sterile filter paper. The plants were then transferred to MS + 0.2 mg·L$^{-1}$ NAA + 1.5 mg·L$^{-1}$ 6-BA + 20 g·L$^{-1}$ sucrose + 20% potato juice + 7 g·L$^{-1}$ agar (PH 5.8) medium. The untreated plants were used as the control; a total of 12 treatment combinations were used, the number of protocorms in each treatment was 300, which were divided into three replicates. The growth status was observed weekly.

### 2.5. Morphological Observations

Eleven months after the colchicine treatment of protocorms, in the ultra-clean bench, 30 healthy and equal-sized plants were randomly selected each from the treatment groups and the control group. The morphological features, including plant height, internode length, stem diameter, and leaf shape index, were determined using vernier calipers, and the data were recorded.

### 2.6. Stomatal Observation

Stomatal measurements were performed on all of the selected plants, and the specific procedure for stomatal observations was as follows:

The leaves of sterile phytopathogenic seedlings of the treatment and control groups were collected, and a slide with a drop of distilled water in the middle was taken. About

3 mm$^2$ of the lower epidermis of the leaves of the treated and control plants was torn with pointed forceps and placed immediately in distilled water on the slide to prevent the stomata from losing water or closing. The coverslip was held on one side with forceps and gently lowered on the distilled water side to prevent the formation of air bubbles. Excess distilled water was aspirated, and the coverslip was pressed on the slide. The slides were observed using an OLYMPUS BX51 microscope (Olympus corporation, Tokyo, JAPAN). The size of guard cells was observed and recorded at 40 $\times$, and the number of stomata in the fixed field of view was counted. A study by Fang et al. [26] on the induction of polyploidy in *Salvia miltiorrhiza* Bge. reported that the plants were considered mutant if the stomatal length diameter of the treated group was $\geq$1.25 times that of the control group.

### 2.7. Chromosomal Observation

The specific procedures for chromosome observation are as follows:

Samples were obtained at around 10:00 a.m. (the time of peak division in the root tip meristem zone of *D. officinale* was determined by performing multiple sampling tests), and young root tips of 1 cm were cut from the treated and control plants. The root tips were immersed in 0.002 mol·L$^{-1}$ of 8-hydroxyquinoline solution and treated at 15–20 °C for about 6 h to obtain more cells in the dividing phase. The pretreated root tips were removed, rinsed thrice with distilled water, immersed in Carnoy's fixative (anhydrous ethanol–glacial acetic acid at a ratio of 3:1, ready to use), and treated for 24 h at 4 °C. The roots should be dissociated immediately or stored temporarily in 70% alcohol for later use. This kills the cells quickly and leaves the cell structure intact. The fixed root tips were removed, rinsed in 90% alcohol, and placed in 1 mol·L$^{-1}$ of HCl for 4 min at 60 °C. This removes the pectin layer between the cells and weakens the cell wall. The dissociated root tips were washed thrice with distilled water, and the white phloem portion of the root tip was cut with a knife, pounded with knife forceps, and stained with modified phenol magenta for about 20 min at room temperature. One side of the coverslip was dipped in the staining solution, excess staining solution was removed with filter paper, and the coverslip was pressed with rubber to disperse the cells in the hyphal zone. The slide was observed using an OLYMPUS BX51 microscope, the number of chromosomes was counted, and the photographic images were obtained.

### 2.8. Chromosome Karyotype Analysis

Karyotypes of diploid and tetraploid *D. officinale* were observed and analyzed according to the features of the microscope system camera. Cells at mid-mitosis with well-dispersed chromosomes and minor overlap in the pressed slides were selected. Chromosomes were paired and measured using Adobe Photoshop software, and the chromosome data were analyzed by Excel software. Then, five cells from each material were taken to obtain the karyotype information, and the average value was taken as the karyotype analysis parameter. The criteria proposed by Li [27] were used for analysis, the classification system proposed by Stebbins was used for the karyotype classification [28], and the karyotype asymmetry coefficient was calculated by referring to the method given by Arano [29].

## 3. Results

### 3.1. Induction of Protocorms of D. officinale

Whitening of the protocorms of each treatment occurred about 2 weeks after colchicine treatment. When the colchicine concentration was kept constant, the number of albino protocorms increased with the increase in the treatment time. When the treatment time was kept constant, the number of albino protocorms increased with the increase in the colchicine concentration (Figure 1). Most albino protoplasts failed to grow, and the few protoplasts that grew to an albino seedling stage also died gradually.

The survival rate of 300 protocorms from each group was determined, which was 99.00% for the control group and within a range of 58.00–91.67% for the treatment groups (Table 1). When both colchicine concentration and treatment time were increased, the

survival rate of the protocorms decreased, the difference between the treatment groups and the control group was also gradually increased, the most significant number of deaths difference was obtained under the treatment with 0.4% colchicine for 48 h. Under the same colchicine concentration, more protocorms died with the increase in the treatment time. Under the same treatment time, the survival rate of the protocorms decreased as the colchicine concentration increased, and after colchicine treatment for 48 h, the difference between the concentrations was more significant. The number of plants in the control group that failed to survive was only three. In the treatment group, the number of plants that failed to survive with the increase in colchicine concentration and treatment time was much higher (i.e., 126) than that in the control group.

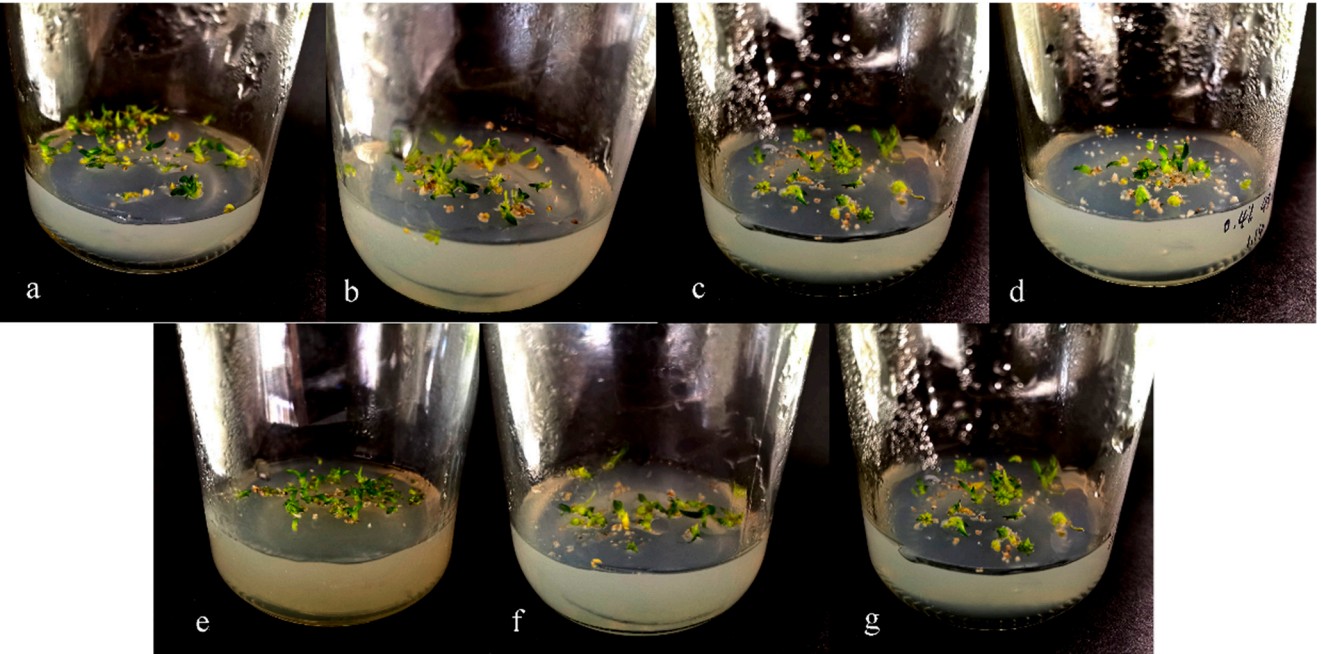

**Figure 1.** Albino phenomenon of protocorm in each colchicine treatment group: (**a**) 0.1% 48 h; (**b**) 0.2% 48 h; (**c**) 0.3% 48 h; (**d**) 0.4% 48 h; (**e**) 0.3% 24 h; (**f**) 0.3% 36 h; (**g**) 0.3% 48 h.

**Table 1.** Effect of colchicine concentration on the survival rate of *Dendrobium officinale* protocorms.

| Processing Time/h | Treatment Concentration/% | Number of Samples | Number of Deaths | Survival Rate/% |
|---|---|---|---|---|
| 0 (CK) | 0 (CK) | 300 | $1.00 \pm 0.58$ h* | 99.00 |
| 24 | 0.1 | 300 | $8.33 \pm 0.67$ g | 91.67 |
| 24 | 0.2 | 300 | $16.67 \pm 0.33$ ef | 83.33 |
| 24 | 0.3 | 300 | $17.33 \pm 1.76$ ef | 82.67 |
| 24 | 0.4 | 300 | $21.00 \pm 0.58$ def | 79.00 |
| 36 | 0.1 | 300 | $14.67 \pm 1.33$ f | 85.33 |
| 36 | 0.2 | 300 | $22.67 \pm 1.20$ de | 77.33 |
| 36 | 0.3 | 300 | $26.33 \pm 0.89$ cd | 74.67 |
| 36 | 0.4 | 300 | $29.67 \pm 1.45$ bc | 70.33 |
| 48 | 0.1 | 300 | $18.33 \pm 1.20$ ef | 81.67 |
| 48 | 0.2 | 300 | $24.00 \pm 0.58$ cde | 76.00 |
| 48 | 0.3 | 300 | $34.33 \pm 2.19$ b | 65.66 |
| 48 | 0.4 | 300 | $42.00 \pm 4.93$ a | 58.00 |

* Different lowercase letters indicate that the difference is significant at $p < 0.05$.

### 3.2. Morphological Observations

The morphologies of the treated and control *D. officials* and the occasional branching growth were observed (Figure 2). Plant height, node spacing, stem diameter, and leaf shape index of all the plants were observed and analyzed (Table 2). The mean plant height of the control group was 22.40 mm, which was higher than that of the treatment groups

(18.17–21.59 mm). The treatment combinations of 0.1% colchicine for 24 h, 0.4% colchicine for 24 h, 0.1% colchicine for 36 h, 0.3% colchicine for 36 h, 0.4% colchicine for 36 h, 0.1% colchicine for 48 h, and 0.3% colchicine for 48 h generated plants with markedly different heights than those of the control group. The most significant height difference was obtained under the treatment with 0.4% colchicine for 36 h, where the mean plant height was only 18.17 mm. The average length of the internodes of the control group was 3.86 mm and that of the treatment groups ranged from 3.14 to 3.99 mm. The treatment combinations of 0.1% colchicine for 36 h, 0.2% colchicine for 36 h, 0.3% colchicine for 36 h, 0.4% colchicine for 36 h, 0.3% colchicine for 48 h, and 0.4% 48 h created plants with markedly different internode lengths compared with that of the control group, and the combination 0.3% colchicine for 48 h created the plant with the most distinct internode length of 3.14 mm. The average stem diameter of the control group was 2.72 mm, the average stem diameter of the treatment groups ranged from 2.85 to 3.45 mm, and the mean stem diameter of the treatment groups treated with 0.1% colchicine for 24 h, 0.2% colchicine for 24 h, and 0.3% colchicine for 24 h was markedly different from that of the control group. The mean leaf shape index of the control group and treatment groups was 1.97 and 1.55–1.78, respectively. The mean leaf shape index of all treatment groups was markedly different from that of the control group; in particular, the mean leaf shape index of the group treated with 0.1% colchicine for 24 h differed most markedly from that of the control group, with a mean leaf shape index of only 1.77.

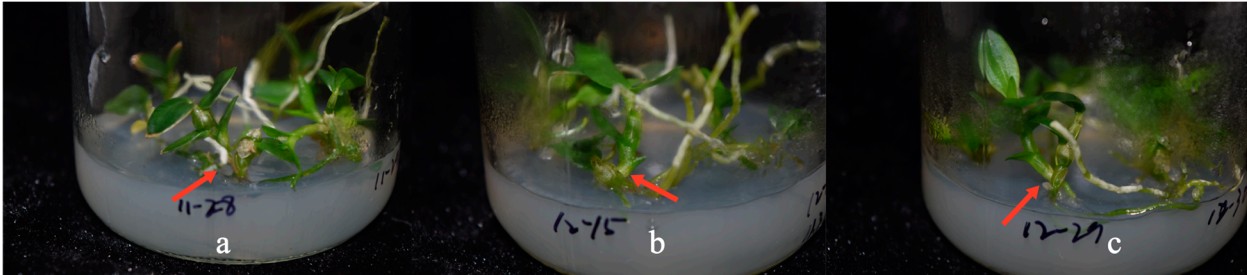

**Figure 2.** Branching growth phenomenon in the colchicine treatment groups: (**a**) 0.3% 48 h; (**b**) 0.4% 48 h; (**c**) 0.4% 48 h. The red arrow showed the branching growth phenomenon.

**Table 2.** Comparison of morphological characteristics of the control plants and plants treated with colchicine.

| Processing Time/h | Treatment Concentration/% | Plant Height/mm | Internodes Length/mm | Stem Diameter/mm | Leaf Shape Index/mm |
|---|---|---|---|---|---|
| 0 (CK) | 0 (CK) | 22.40 ± 0.77 a* | 3.86 ± 0.09 a | 2.72 ± 0.09 d | 1.97 ± 0.08 a |
| 24 | 0.1 | 18.97 ± 0.91 bcd | 3.98 ± 0.19 a | 3.45 ± 0.9 a | 1.77 ± 0.04 bc |
| 24 | 0.2 | 21.27 ± 0.99 abc | 3.89 ± 0.11 a | 3.37 ± 0.10 b | 1.55 ± 0.05 d |
| 24 | 0.3 | 21.59 ± 1.08 ab | 3.99 ± 0.13 a | 3.32 ± 0.12 bc | 1.61 ± 0.05 cd |
| 24 | 0.4 | 18.27 ± 0.70 cd | 3.82 ± 0.10 ab | 2.97 ± 0.08 bcd | 1.77 ± 0.05 bcd |
| 36 | 0.1 | 18.79 ± 0.81 bcd | 3.60 ± 0.07 bcd | 3.24 ± 0.33 bcd | 1.66 ± 0.04 bcd |
| 36 | 0.2 | 20.46 ± 0.95 abcd | 3.50 ± 0.09 bcd | 2.91 ± 0.09 cd | 1.78 ± 0.07 bc |
| 36 | 0.3 | 18.88 ± 0.66 bcd | 3.45 ± 0.07 bcd | 2.88 ± 0.09 cd | 1.77 ± 0.08 bc |
| 36 | 0.4 | 18.17 ± 0.87 d | 3.35 ± 0.09 cd | 2.85 ± 0.21 cd | 1.77 ± 0.05 bc |
| 48 | 0.1 | 19.07 ± 0.73 bcd | 3.67 ± 0.09 abc | 2.95 ± 0.09 bcd | 1.73 ± 0.04 bc |
| 48 | 0.2 | 20.62 ± 1.17 abcd | 3.91 ± 0.10 ab | 3.06 ± 0.12 bcd | 1.79 ± 0.07 b |
| 48 | 0.3 | 18.80 ± 0.88 bcd | 3.14 ± 0.05 d | 2.98 ± 0.09 d | 1.62 ± 0.04 cd |
| 48 | 0.4 | 20.29 ± 0.95 abcd | 3.28 ± 0.08 cd | 2.98 ± 0.08 bcd | 1.64 ± 0.06 bcd |

* Different lowercase letters indicate that the difference is significant at $p < 0.05$.

In addition, to further screen for polyploid phenotypic markers, we compared the morphological indicators of tetraploid and control diploid plants based on chromosomal observations (Table 3). The mean plant height was 24.72 mm for diploid plants and 21.47 mm for tetraploid plants, indicating a difference of 13.14% between them, which was nonsignificant. The mean internode length was 3.98 mm and 3.36 mm for diploids and

tetraploids, respectively, indicating a 7.69% reduction in the internode length of tetraploids. The mean stem diameter was 2.53 mm for the diploids and 3.27 mm for the tetraploids, indicating a difference of 29.25% between them, which was significant. The difference in mean leaf width was not significant. The mean length of the tetraploid plant leaves was 1.5 mm less than that of the diploid plant leaves, indicating a 0.12-fold reduction in length. Accordingly, the tetraploid leaf shape index (1.79) was lower than the diploid leaf shape index (2.32).

**Table 3.** Comparison of morphological characteristics of diploid and tetraploid plants.

| Morphological Characteristics | Diploid Plants | Tetraploid Plants |
| --- | --- | --- |
| Plant height/mm | 22.40 ± 0.77 b* | 21.47 ± 1.67 b |
| Internodes length/mm | 3.86 ± 0.09 a | 3.36 ± 0.17 b |
| Stem diameter/mm | 2.72 ± 0.09 b | 3.27 ± 0.18 a |
| Leaf shape index | 1.97 ± 0.08 a | 1.79 ± 0.75 b |

* Different lowercase letters indicate that the difference is significant at $p < 0.05$.

### 3.3. Stomatometric Observations

The number of stomata of *D. officinale* in the treatment and control groups was counted, and their lengths and diameters were measured. The variation rate in the treatment groups was obtained by comparing the stomatal data of the treatment groups with those of the control group (Table 4). The observation and analysis of 30 stomata in each group showed the variation in the treatment groups. Under the treatment with 0.4% colchicine for 36 h, the variation rate was the lowest (6.67%). The variation rate of the combinations 0.1% for 24 h, 0.2% for 24 h, 0.3% for 24 h, and 0.1% for 48 h exceeded 20.00%. Under the treatment with 0.3% colchicine concentration for 24 h, the variation rate increased to 30.00%.

**Table 4.** Plant variation and doubling statistics for the colchicine treatment groups.

| Processing Time/h | Treatment Concentration/% | Number of Samples | Stomatal Variation Rate/% | Number of Chromosome Doubles |
| --- | --- | --- | --- | --- |
| 0 (CK) | 0 (CK) | 30 | 0.00 | 0 |
| 24 | 0.1 | 30 | 23.33 | 0 |
| 24 | 0.2 | 30 | 26.76 | 1 |
| 24 | 0.3 | 30 | 30.00 | 1 |
| 24 | 0.4 | 30 | 20.00 | 0 |
| 36 | 0.1 | 30 | 10.00 | 0 |
| 36 | 0.2 | 30 | 20.00 | 1 |
| 36 | 0.3 | 30 | 13.33 | 1 |
| 36 | 0.4 | 30 | 6.67 | 0 |
| 48 | 0.1 | 30 | 26.67 | 0 |
| 48 | 0.2 | 30 | 13.33 | 3 |
| 48 | 0.3 | 30 | 13.33 | 2 |
| 48 | 0.4 | 30 | 16.67 | 1 |

To further identify whether stomatal variability could be used as a marker for polyploidy induction, we also comparatively analyzed the stomatal measurement data of tetraploid and diploid plants, which were determined based on chromosome observations. We found that the stomata of the diploid plants were mostly ovoid and of equal size, whereas the stomata of the tetraploid plants were suborbicular with slightly different sizes (Figure 3). The average density of stomata in the lower epidermis of the diploid and tetraploid leaves under the same field of view was 10 and 6 stomata/mm$^2$, respectively, indicating a significant variation in the number of stomata per unit area (Table 5). The average length and short diameter of the stomata of the tetraploid plants were 53.50 μm and 48.13 μm, respectively, which were 40.46% and 23.92% higher than those of the stomata of the diploid plants, respectively.

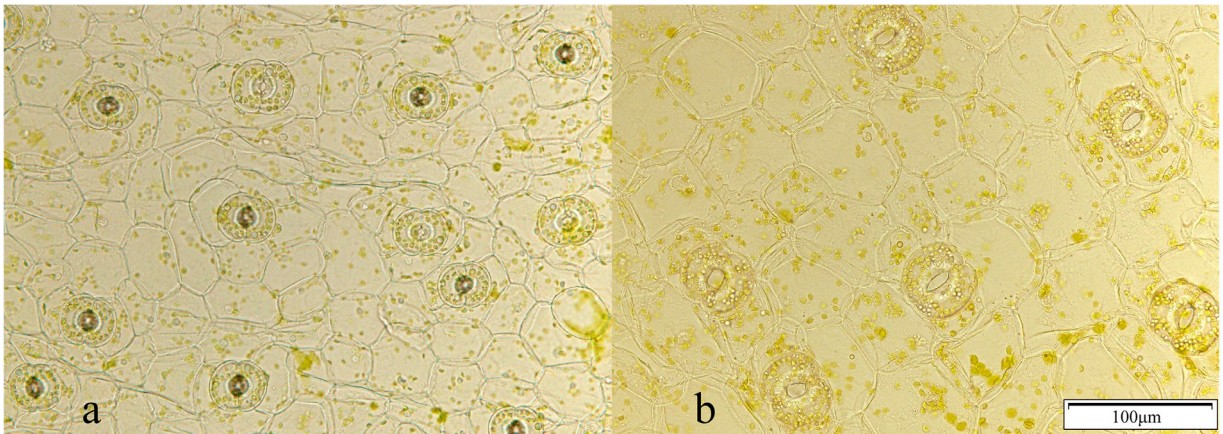

**Figure 3.** Stomata of *Dendrobium officinale* (40×): (**a**) diploid plants; (**b**) tetraploid plants.

**Table 5.** Comparative stomatal characteristics of *Dendrobium officinale*.

| Stomatal Indicators | Diploid Plants | Tetraploid Plants |
| --- | --- | --- |
| Stomatal length diameter/μm | 38.09 ± 0.94 b* | 53.50 ± 2.86 a |
| Stomatal short diameter/μm | 34.84 ± 0.79 b | 48.13 ± 2.33 a |
| Stomatal density/mm$^2$ | 9.56 ± 0.38 a | 6.00 ± 0.41 b |

* Different lowercase letters indicate that the difference is significant at $p < 0.05$.

### 3.4. Chromosome Observations

The number of chromosomes was 2n = 2x = 38 for diploid *D. officinale* and 2n = 4x = 76 for tetraploid *D. officinale* (Figure 4). A total of 10 tetraploid *D. officinale* plants were successfully obtained from colchicine-induced *D. officinale* protoplasts. The highest induction rate for the tetraploid plants was 10.00%, and the induction condition was 0.2% colchicine treatment for 48 h. Three tetraploids were obtained after induction (Table 4).

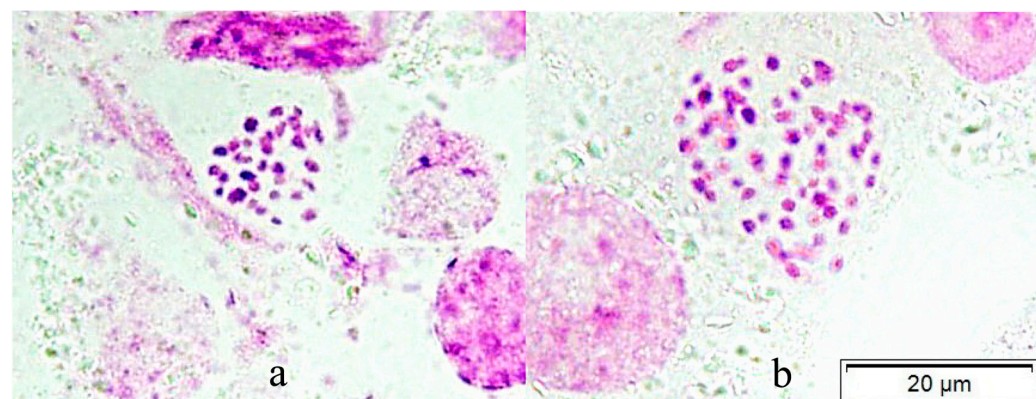

**Figure 4.** Chromosomes of *Dendrobium officinale* (100×) (**a**) diploid plants and (**b**) tetraploid plants.

### 3.5. Chromosome Karyotype Analysis

Karyotype analysis was performed on diploid and tetraploid *Dendrobium* chromosomes at the mid-division phase, and the chromosomes were measured, paired, and aligned to produce the morphological maps (Figure 5), namely the karyotype map (Figure 6) and karyotype pattern map (Figure 7), of mid-division chromosomes. The number of chromosomes in diploid *D. officinale* was 38, with a chromosome base of x = 19. After doubling, the chromosome base of tetraploid *D. officinale* remained unchanged, and the number of chromosomes increased to 76.

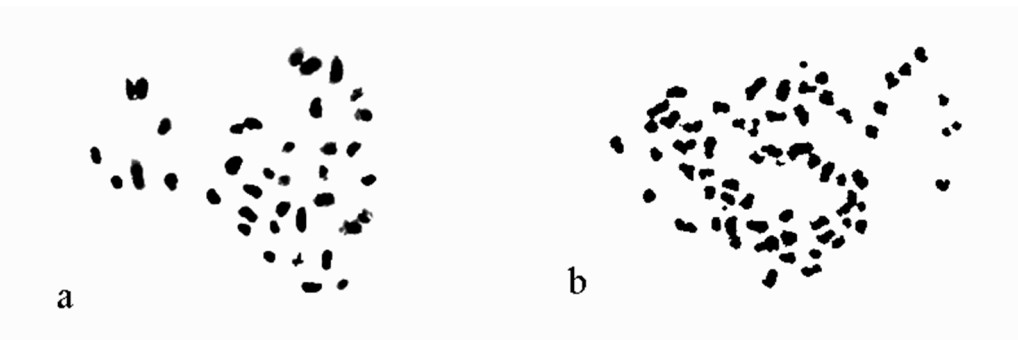

**Figure 5.** Chromosome morphology of *Dendrobium officinale:* (**a**) diploid plants; (**b**) tetraploid plants.

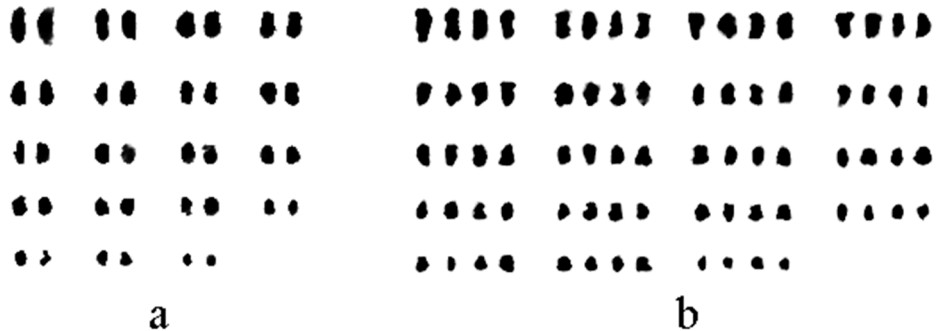

**Figure 6.** Karyotype diagram of *Dendrobium officinale:* (**a**) diploid plants; (**b**) tetraploid plants.

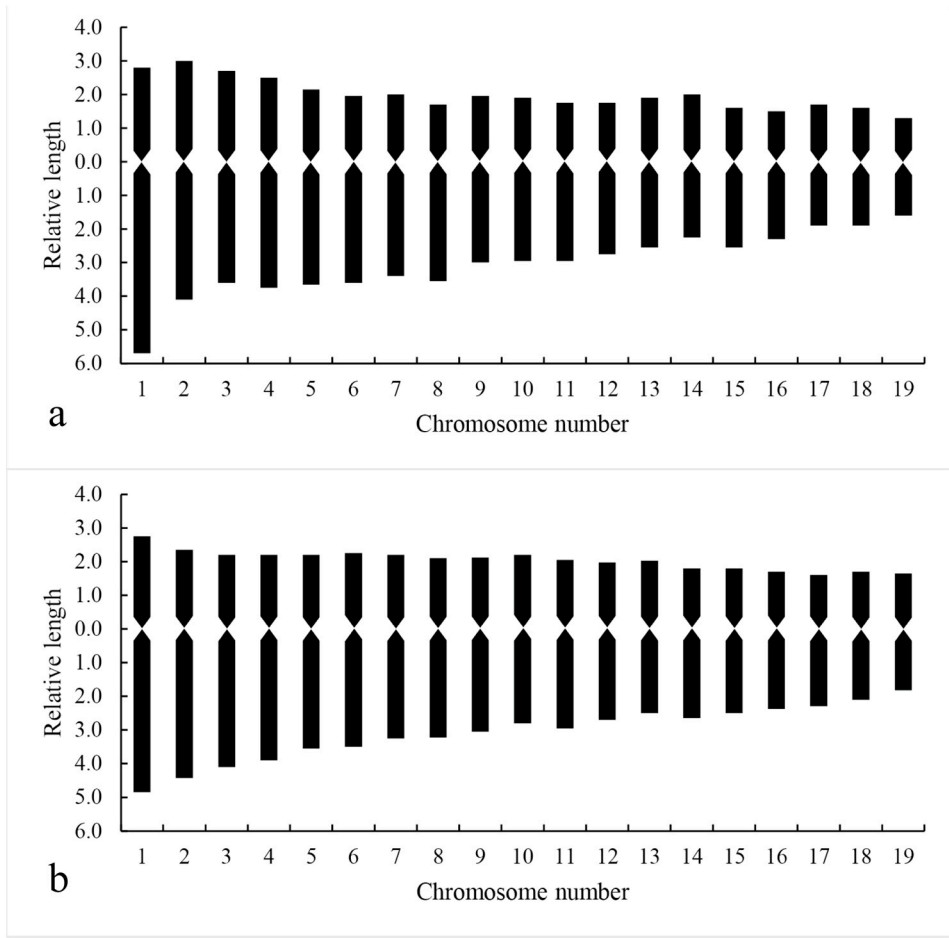

**Figure 7.** Karyotype pattern diagram of *Dendrobium officinale:* (**a**) diploid plants; (**b**) tetraploid plants.

The diploid karyotype formula was 2n = 2x = 38 = 24m + 14sm, with a karyotype asymmetry coefficient of 60.59%, and the proportion of chromosomes with an arm ratio of >2 was 36.84%, accounting for type 2B. The chromosome composition of diploids is of two types, m and sm. The relative chromosome lengths ranged from 2.9 to 8.5, with an average arm ratio of 1.54, arm ratio range of 1.10–2.15, and length ratio for the longest to the shortest chromosome of 2.93.

The tetraploid karyotype formula was 2n = 4x = 76 = 58m + 18sm, with a karyotype asymmetry coefficient of 60.04%, and the proportion of chromosomes with an arm ratio of >2 was 23.68%, accounting for type 2B. The chromosome composition of tetraploids is of two types, m and sm. The relative lengths ranged from 3.2 to 8.1, with an average arm ratio of 1.49, arm ratio range of 1.06–2.2, and length ratio for the longest to the shortest chromosome of 2.53.

The relative chromosome length ranges for diploid and tetraploid *D. officinale* were 2.9–8.5 and 3.2–8.1, respectively (Table 6), and the relative chromosome length range was smaller for the tetraploids. The arm ranges for the diploid and tetraploid plants were 1.10–2.15 and 1.06–2.2, respectively, indicating a higher range of arm ratios for tetraploids. The tetraploid *D. officinale* karyotype type still consisted of two types, m and sm. The diploid chromosome karyotype was of the sm type, accounting for 36.84%, whereas the percentage of the tetraploid chromosome karyotype, the sm type, decreased to 23.68%. After diploid doubling, the karyotype category was still 2B, and the asymmetry coefficient decreased.

**Table 6.** Karyotype comparison of diploid and tetraploid *Dendrobium officinale*.

| Ploidy | Relative Length | Arm Ratio | Length Ratio of Longest Chromosome to Shortest Chromosome | Chromosomes with Arm Ratio > 2/% | Karyotype Formula | Karyotype | Asymmetry Coefficient/% |
|---|---|---|---|---|---|---|---|
| Diploid plants | 2.9~8.5 | 1.10~2.15 | 2.93 | 36.84 | 2n = 2x = 38 = 24m + 14sm | 2B | 60.59 |
| Tetraploid plants | 3.2~8.1 | 1.06~2.2 | 2.53 | 23.68 | 2n = 4x = 76 = 58m + 18sm | 2B | 60.04 |

## 4. Discussion

### 4.1. Polyploid Protocorm Induction of D. officinale

Colchicine could affect the cellular tissues of the plant, with the treatment groups showing bleaching of the protocorms and slow growth of the plants that survived. Wu [19] also noted the phenomenon of bleaching in a study on colchicine-induced *D. officinale* protoplasts. Zhang [17] treated *D. officinale* seed embryos with colchicine and observed different degrees of growth retardation, which is consistent with the results of this study.

The survival rate of the protocorms in the treatment groups decreased with the increase in colchicine concentration and treatment time. The highest number of deaths occurred under the treatment with 0.4% concentration for 48 h. Therefore, the survival rate of the treated primary bulbs was correlated with the colchicine concentration and treatment time, with higher concentrations and longer treatment times resulting in lower survival rates. Vichiato et al. [30] performed the polyploidization induction of *D. officinale* and found that the colchicine concentration and treatment time were inversely related to the survival rate. A study on the effects of colchicine on *Dendrocalumus brandisii* (Munro) Kurz [31] seeds and seedlings showed that seed germination and seedling growth decreased continuously with the increasing colchicine concentration. The pattern of changes in the survival rate observed in this study was consistent with that of explants reported in a previous [32] polyploidization study.

### 4.2. Morphological Observations

Based on the morphological observations, some plants in the treatment showed a decreased plant height and shortened internode length; however, the overall mean value did not change markedly. Significant changes in the mean values of the stem diameter

and leaf shape index were observed in both the treatment and control groups. The mean value of the stem diameter in the treatment groups increased significantly by 4.78–41.91%, whereas the mean value of the leaf shape index in the treatment groups decreased by 9.65–21.32%.

The morphological data analysis of diploid and tetraploid *D. officinale* showed that the difference in plant height between the diploids and tetraploids was not significant. In contrast, the differences in internode length and stem diameter between the diploids and tetraploids were significant. The leaf shape index decreased in the tetraploids compared with that in the diploids. These results are consistent with those of previous studies [19,33]. Moreover, no obvious change in leaf color and no morphological variation in the leaves were observed in this study. In a study on the induction of polyploids in *D. officinale*, Liao [34] found that the leaves of mutant plants were rough and wrinkled with deepened veins. Zhan et al. [35] also found that the *D. officinale* mutant had serrated leaves and distorted leaf shapes, which might be because of differences in the source and quality of the seeds of the mutagenic material. In addition, we found individual branches of *D. officinale* in the treatment group, and this unexpected morphological feature will be further observed and explored.

*4.3. Stomatometric Observations*

Based on the stomatal morphometric measurements, plants in the treatment groups showed changes such as a decreased number of stomata per unit area, increased length and diameter of stomata, and stomatal deformation. The results showed that the variation rate did not increase with increasing the concentration and time; plants treated with 0.3% colchicine for 24 h showed variation up to 30.00%. Based on the stomatal morphometric data of diploid and tetraploid *D. officinale*, the polyploid plants showed a significant increase in the number of leaf stomata and a decrease in densities. However, based on chromosome observations, we found that the treatment combinations at the highest tetraploid induction rate differed from those at the highest variation rate. These typical characteristics can be helpful for ploidy identification by combining multiple methods to narrow down the range of chromosome observations and improve the efficiency of ploidy identification. However, they cannot be used directly for the identification of polyploids as an alternative to chromosome observation. After the plant cell chromosomes were doubled, the size and density of the epidermal leaf stomata and the size of guard cells in the tetraploid plants varied significantly from those in the diploid plants [36]. Li et al. [37] compared diploid lines of *Echinacea purpurea* L. with polyploid plants and found that tetraploid and octoploid plants showed more retarded growth and development, larger stomata, lower stomatal frequency, and a higher number of chloroplasts in the guard cells. Tokumoto et al. [38] used colchicine to induce *Eucommia ulmoides* Oliver to obtain tetraploid plants. The comparison showed that the tetraploids had larger stomatal sizes and lower stomatal densities than the diploids. These results are consistent with those of previous studies.

*4.4. Chromosome Observations*

The chromosome observation results showed that the highest induction rate was 10.00% in the tetraploid plants, which was achieved under an induction condition of 0.2% colchicine treatment for 48 h. In this experiment, the chromosomes of *D. officinale* were counted using the conventional pressing method, and after pressing the root tips, the chromosome number of diploid *D. officinale* was $2n = 2x = 38$, as observed under the microscope. The chromosome number of *D. officinale* mutants treated with colchicine was $2n = 4x = 76$, and that in some of the mutants was doubled. Furthermore, the mutation rate did not vary with increasing the colchicine concentration and treatment time. A similar phenomenon was reported in a study on *Populus alba* L. polyploidy by Ren et al. [33], which is in agreement with the results of a study on the polyploidization of *Lycium ruthenium* Murr. performed by Gao et al. [39].

*D. officinale* grows slowly, and the root tips show aerial growth. The newly grown root tips are tender, green, short, and not easy to obtain, and the chromosomes are short and abundant in number; thus, obtaining chromosomes at the division phase is difficult [19]. After many trials, we found that cell division in *D. officinale* root tips was more vigorous from 09:30 to 10:00 a.m., and more chromosomes at the division phase were obtained from the intercepted root tips during this period. After dissociation, the root tips were properly rinsed with sterile water to remove cellulose and pectin from the cell wall for better staining results. Liao [34] performed an in-depth study on the pretreatment, fixation, and dissociation of *D. officinale* root tip compression and found that the sampling time for the *D. officinale* root tips was from 08:00 to 09:00 a.m., which is different from that in the present study. Wu [19] reported a sampling time of 09:30–10:30, which is similar to that in our study. The root tip sampling time was slightly different, probably because of the use of different materials. However, *Dendrobium* root tip chromosome observation needs further exploration to improve chromosome photographic imaging.

Many methods are available for the identification of polyploid induction, of which chromosome counting is the most reliable method. Conventional chromosome pressing is complex, technically demanding, and time consuming [34]. In the present study, some of the treated plants did not double their chromosomes, but showed morphological or stomatal variations, such as slower growth, shorter internodes, fewer stomata, and larger stomata, which might have been caused by colchicine treatment. Ploidy level was measured in annual *D. officinale*, and no specific indicator was found to be associated with plant ploidy, probably because of the slow growth of *D. officinale* and the nonsignificant association of phenotypic data with ploidy. Therefore, exploring other measurement indicators that strongly correlate with ploidy may improve the efficiency of ploidy identification.

*4.5. Karyotype Analysis of Chromosome Polyploidy*

The present study showed that the metaphase chromosomes of diploid and tetraploid *D. officinale* were mainly of the m and sm types, and no satellite chromosomes were found. Furthermore, the symmetry of karyotypes was generally neat. The number of karyotypes within a species is m-type > sm-type [40], and the number of m chromosomes is greater than that of the sm chromosomes. The present study showed similar findings. The karyotype formula for diploid *D. officinale* was $2n = 2x = 38 = 24m + 14sm$ and that for the tetraploid karyotype was $2n = 4x = 76 = 58m + 18sm$, with the karyotype category 2B, and the asymmetry coefficient of tetraploids was lower than that of the diploids. The karyotype formula of tetraploid *D. officinale* did not increase exponentially, where the number of m-type chromosomes was more than twice the number of original diploid m-type chromosomes, but the number of sm-type chromosomes was not twice the number of original diploid sm-type chromosomes. This may be because of the following reasons: (1) the chromosome type changed after doubling because of the effect of colchicine treatment, or (2) the sm-type chromosome was not doubled entirely, whereas the m-type chromosome was doubled greatly, resulting in the karyotype formula not increasing exponentially.

The karyotype formula of tetraploid *L. ruthenium*. developed by Gao et al. [39] is basically the same as that of diploids, with the karyotype type remaining changed. Samatadze et al. [41] performed a karyotype analysis of diploid and tetraploid *P. caeruleum* and showed that the karyotype of tetraploid *P. caeruleum* plants contained two similar genomes. The karyotype formulas and karyotypes of the artificially induced polyploids showed a quantitative increase in the ploidy level compared with those of the diploid plants, whereas none of the karyotypes changed. In the present study, the karyotype formula ratio of *D. officinale* showed changes in diploids and tetraploids, which is not consistent with the results of previous polyploidy studies; however, the karyotype type of the tetraploids was the same as that of the diploids, which is consistent with the patterns observed in previous studies. As the *D. officinale* chromosomes were small and numerous, determining the quality of the press and arm length was difficult, which would affect the results of the karyotype analysis,

or it may also be caused by the chromosome specificity of *D. officinale* or *Dendrobium* and even the orchid family. However, the present study results need to be further verified.

Identifying chromosomal variations is crucial for the evolutionary analysis, research, and breeding of new cultivars. Karyotypic parameters are one of the crucial indicators for classifying and identifying plants, thus enabling the study of their genetic diversity in cytological terms [42]. Scholars have also explored polyploidy in natural evolution. Noedoost et al. [43] performed a karyotype analysis of existing *Chara* species of different ploidies and reported different karyotype formula ratios and karyotype types. The karyotype analysis of *Hemerocallis* of different ploidy levels was performed by Zhang et al. [44], and although different materials of the same ploidy level showed considerable differences, the proportion of karyotype formulas and karyotype types differed across different ploidy levels. Li et al. [45] performed a karyotype analysis of diploid, tetraploid, and hexaploid *Camellia sinensis* plants and found different karyotype formula ratios and karyotype types. In a study on wild roses by Yu et al. [46], diploids and tetraploids were found in different floras. The tetraploid karyotype had the highest asymmetry index and the most evolved arm ratio among all of the taxa. Therefore, heterozygosity and polymorphism exist in polyploids that have evolved over a long period in nature [47], thus leading to great variations in their karyotype formulas and karyotype categories. In contrast, artificially induced polyploids exhibit less or no changes in karyotype formulas and karyotype types compared with diploids. This may be because artificially induced polyploids, which tend to simply double their chromosome number, have not undergone long-term evolution and natural selection.

*4.6. Polyploidy Effects on Gene Expression*

Polyploidy has considerable effects on duplicate gene expression, including silencing and up- or down-regulation of one of the duplicated genes [48]. Mishra et al. [49] found that expression analysis through various known genes involved in the biosynthesis of morphinanes in diploids and tetraploids showed an increased expression in *Papaver somniferum* L. tetraploids. On the other hand, many changes in gene expression may occur only in a specific organ, so the relative expression of duplicated genes can vary in different parts of the plant [12]. The expression of alkaloid-biosynthesis-related genes in leaves and roots of induced tetraploid of *Papaver bracteatum* Lindl compared with its diploid plants was analyzed [50], and showed that the expression of berberine bridging enzyme (BBE) in the leaves was significantly decreased, while it was increased in the root tissues. In the study of the gene level of *D. officinale*, Wu [19] investigated the content of polysaccharide in the identified tetraploid and diploid plants, and analyzed the difference in gene expression of polysaccharide-synthesis-related genes (PEPC, UGP, and SPS) in diploid and tetraploid plants, and found that all the content of polysaccharides and the expression of those genes increased in tetraploid plants. According to the current research, the composition and content of polysaccharide in the identified tetraploid and diploid should be further analyzed and identified. The differential expression genes in different tissues between the diploid and tetraploid of *D. officinale* need further investigation, especially those genes related to the biosynthesis of polysaccharides.

**5. Conclusions**

In this study, 10 *D. officinale* tetraploid plants were successfully cultivated by inducing the protocorms of *D. officinale* with different colchicine concentrations, and the optimum treatment condition for inducing tetraploid *D. officinale* was determined. The expression of the variation rate and tetraploid induction rate was standardized, which provided a clear reference value for future ploidy breeding research. The karyotype formulas of diploid and tetraploid *D. officinale* were derived from the karyotype analysis to determine the karyotype categories. These findings provide a basis for improving *D. officinale* polyploid cultivars and are valuable for the research on *D. officinale* polyploids and the selection and breeding of *D. officinale* varieties.

**Author Contributions:** Conceptualization, S.-C.N.; methodology, Y.J.; software, Y.L.; validation, Y.L. and S.-D.D.; investigation, Y.L.; resources, L.-H.H. and D.-Y.X.; data curation, Y.L.; writing—original draft preparation, Y.L.; writing—review and editing, S.-C.N., D.-F.C., L.-H.H. and D.-Y.X.; visualization, Y.L. All authors have read and agreed to the published version of the manuscript.

**Funding:** This work was supported by the Young Talent Project of Hebei Agricultural University Foundation (YJ201848) and the Funds of the Hebei Province Natural Science Foundation (C2019204295, C2022204214).

**Data Availability Statement:** Not applicable.

**Conflicts of Interest:** The authors declare no conflict of interest.

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
