# Peer review of "Polyploid Induction and Karyotype Analysis of Dendrobium officinale"

_horticulturae, doi:10.3390/horticulturae9030329_

Round 1

Reviewer 1 Report

The current version of manuscript have interesting results. The paper needs revision. I have few specific comments.

1)    The contribution of the current study to horticultural research should be identified and written in the conclusion section.

2)    Author should be concise/short in discussion section, especially 4.2.

3)    How the polyploid induction affects at gene/allele level? Please discuss

4)    Check English. Line 44-45

5)    Throughout the manuscript editing for English language is required. Please consider revising whole Introduction section.

6)    How colchicine concentration determined?

7)    Downfall of current study? Future direction? Unexpected results? Include in discussion

Author Response

Responses to Reviewers’ Comments

Dear Editor,

Thank you for your suggestive comments on our manuscript entitled “Polyploid Induction and Karyotype Analysis of Dendrobium Officinale” (Manuscript ID: horticulturae-2215864). We have now revised our manuscript accordingly. Changes made in response to your comments are detailed below (our responses are in blue).

With best regards,

Shan-Ce Niu (on behalf of all authors)

Reviewer #1:

Comments:
1)    The contribution of the current study to horticultural research should be identified and written in the conclusion section.

Response:  

Thank you for spending time reviewing our manuscript and providing us with constructive comments. we have added a summary of horticultural research in the conclusion, as suggested by the reviewers.

2)    Author should be concise/short in discussion section, especially 4.2.

Response:

We have revised the discussion section in the revised version of the manuscript.

3)    How the polyploid induction affects at gene/allele level? Please discuss

Response:

In the revised version of our manuscript, we have increased the effect of polyploid induction on the gene level in the discussion section.

4)    Check English. Line 44-45

Response:

We have revised the manuscript to make it better for reading, not only in language but also in logic.

5)    Throughout the manuscript editing for English language is required. Please consider revising whole Introduction section.

Response:

We have revised the manuscript to make it better for reading, not only in language but also in logic. We have the revised version of our manuscript edited for English language by TopEdit, with regard to grammar, punctuation, spelling and clarity. Their language editors are native English speakers with long-term experience in editing scientific and technical manuscripts.

In the revised version of our manuscript, we have made some supplements and modifications in the introduction.

6)    How colchicine concentration determined?

Response:

This concentration is determined by referring to the treatment concentration of previous studies. As suggested by the reviewer, we have marked the manuscript for better clear material and method sections.

7) Downfall of current study? Future direction? Unexpected results? Include in discussion

Response:

We have increased downfall of current study, future directions, unexpected results in the revised manuscript. The revised part has been marked yellow in the revised manuscript.

For more specific corrections, please see the revised manuscript.

Reviewer 2 Report

The manuscript is focused on polyploid induction in Dendrobium officinale by treating the protocorms with colchicine. Studies on this issue have already been performed (cited in this paper), therefore the study lacks novelty. Colchicine is a well-known mutagen used for polyploidization and the performed here analyses of polyploid detection, chromosome counting, and karyotyping are old.

Some other issues include:

Please provide the full botanical name of the species studied - Dendrobium officinale Kimura ex Migo.

Line 22: induction of what?

Throughout the text, the authors are mixing the terms variety and cultivar, as well as sterilization and disinfection.

Keywords should be arranged alphabetically and should not repeat words from the title.

The correct abbreviation is et al. (a dot is missing).

Lines 82-83: grammar and punctuation.

The unit style is incorrect (it should be mg·L-1, not mg/L).

The materials and methods section is incomplete. What culture vessel was used? What medium was used for germination? How much medium? How many seeds per vessel? What were the physical conditions in the growth room. Please provide details on the producers of key chemicals and equipment used, including colchicine (name, city, state, and country). How old were the plants included in the morphological observations?

Some parts of the Results are unclear, e.g. line 185.

Table 1 lacks statistical analysis.

Captions to figures and titles of tables need to be clarified.

Figure 2: It is unclear which treatments are shown in each photograph.

Line 274: This is not a result. Avoid repeating Materials and methods.

Figure 4: Poor quality of the images.

Sometimes you are referring to full botanical names of species and sometimes not. In general, the full botanical name of the species should be provided when mentioned for the first time.

Line 445: Species or cultivars?

MDPI uses a serial comma.

References lack DOI numbers.

For more specific comments, please see the corrected manuscript.

Author Response

Responses to Reviewers’ Comments

Dear Editor,

Thank you for your suggestive comments on our manuscript entitled “Polyploid Induction and Karyotype Analysis of Dendrobium Officinale” (Manuscript ID: horticulturae-2215864). We have now revised our manuscript accordingly. Changes made in response to your comments are detailed below (our responses are in blue).

With best regards,

Shan-Ce Niu (on behalf of all authors)

Reviewer #2:

Comment 1:

The manuscript is focused on polyploid induction in Dendrobium officinale by treating the protocorms with colchicine. Studies on this issue have already been performed (cited in this paper), therefore the study lacks novelty. Colchicine is a well-known mutagen used for polyploidization and the performed here analyses of polyploid detection, chromosome counting, and karyotyping are old.

Response:

Thank you for spending time reviewing our manuscript and providing us with constructive comments.

As per previous studies of polyploidization, the criteria for determining plant variation are different based on plant morphological differences, plant stomatal differences, and chromosomal numbers. However, different determination criteria yielded different rates of polyploidy, making it difficult to provide an important reference for further research. In this study, the variation rate and induction rate were determined on the basis of the stomatal differences of the treated plants and the chromosome doubling number, respectively.

Comment 2:

Some other issues include:

1) Please provide the full botanical name of the species studied Dendrobium officinale Kimura ex Migo.

Response:

In the revised version of our manuscript, we have made revised.

2) Line 22: induction of what?

Response:

We have clarified this part in the revised manuscript.

3) Throughout the text, the authors are mixing the terms variety and cultivar, as well as sterilization and disinfection.

Response:

We have clarified this part in the revised manuscript.

4) Keywords should be arranged alphabetically and should not repeat words from the title.

Response:

We have revised the keywords in the revised manuscript.

5) The correct abbreviation is et al. (a dot is missing).

Response:

In the revised version of our manuscript, we have made revised.

6) Lines 82-83: grammar and punctuation.

Response:

We have revised this part grammar and punctuation in the revised manuscript.

7) The unit style is incorrect (it should be mg·L-1, not mg/L).

Response:

We have revised the unit style in the revised manuscript.

8) The materials and methods section is incomplete. What culture vessel was used? What medium was used for germination? How much medium? How many seeds per vessel? What were the physical conditions in the growth room. Please provide details on the producers of key chemicals and equipment used, including colchicine (name, city, state, and country). How old were the plants included in the morphological observations?

Response:

We have revised the keywords in the revised manuscript.

The information of culture vessels, culture medium, culture conditions, drug information, plant sampling stage and status has been improved. Protocorm culture was used to provide materials for colchicine treatment, so seeds were not counted, but protocorms were counted when protocorms were treated with colchicine, this information is also added to the revised manuscript.

9) Some parts of the Results are unclear, e.g. line 185.

Response:

We have clarified this part in the revised manuscript.

10) Table 1 lacks statistical analysis.

Response:

We have added data statistical analysis to this part in the revised manuscript.

11) Captions to figures and titles of tables need to be clarified.

Response:

In the revised version of our manuscript, we have made revised to the captions to figures and titles of tables.

12) Figure 2: It is unclear which treatments are shown in each photograph.

Response:

We have clarified the information of each photograph in the revised manuscript.

13) Line 274: This is not a result. Avoid repeating Materials and methods.

Response:

We have revised and deleted this part in the revised manuscript.

14) Figure 4: Poor quality of the images.

Response:

In the revised version of our manuscript, we have selected new images to improve images quality.

15) Sometimes you are referring to full botanical names of species and sometimes not. In general, the full botanical name of the species should be provided when mentioned for the first time.

Response:

We have clarified this part in the revised manuscript.

16) Line 445: Species or cultivars?

Response:

We have clarified the corresponding part in the revised manuscript.

17) MDPI uses a serial comma.

Response:

We have revised the corresponding part in the revised manuscript.

18) References lack DOI numbers.

Response:

We have added the DOI numbers in the revised manuscript.

For more specific corrections, please see the revised manuscript.

Round 2

Reviewer 2 Report

Lux should be converted to PPFD.
